# Sugar Molecules Detection via C_2_N Transistor-Based Sensor: First Principles Modeling

**DOI:** 10.3390/nano13040700

**Published:** 2023-02-11

**Authors:** Asma Wasfi, Sarah Awwad, Mousa Hussein, Falah Awwad

**Affiliations:** 1Department of Electrical and Communication Engineering, College of Engineering, United Arab Emirates University, Al Ain P.O. Box 15551, United Arab Emirates; 2Specialized Rehabilitation Hospital, Abu Dhabi, United Arab Emirates

**Keywords:** C_2_N, first-principles, glucose sensor, nitrogenated holey graphene, non-equilibrium green’s function (NEGF)

## Abstract

Real-time detection of sugar molecules is critical for preventing and monitoring diabetes and for food quality evaluation. In this article, a field effect transistor (FET) based on two-dimensional nitrogenated holey graphene (C_2_N) was designed, developed, and tested to identify the sugar molecules including xylose, fructose, and glucose. Both density functional theory and non-equilibrium Green’s function (DFT + NEGF) were used to study the designed device. Several electronic characteristics were studied, including work function, density of states, electrical current, and transmission spectrum. The proposed sensor is made of a pair of gold electrodes joint through a channel of C_2_N and a gate was placed underneath the channel. The C_2_N monolayer distinctive characteristics are promising for glucose sensors to detect blood sugar and for sugar molecules sensors to evaluate food quality. The electronic transport characteristics of the sensor resulted in a unique signature for each of the sugar molecules. This proposed work suggests that the developed C_2_N transistor-based sensor could detect sugar molecules with high accuracy.

## 1. Introduction

Carbohydrates are important organic substances for both people and plants because they play a number of crucial roles in growth and development. Glucose and fructose have significant importance since they are important nutrients in people diet. Moreover, xylose levels are measured to check if there is problem with peoples’ ability to absorb nutrients. They can be found naturally in a variety of foods or additives. The detection of these sugar molecules is highly important to evaluate food quality [1,2]. As well, reliable and quick sugar detection during food production and storage is highly important. The detection of glucose, fructose, and xylose can be utilized to assess food quality since they reveal details about a food product’s nutritional value, flavor, and sweetness.

Simple sugars such as glucose are frequently present in fruits, vegetables, and grains. High glucose levels in a food product can be a sign that it is high in carbohydrates and energy, but they can also be a sign that the food is overripe or that it has been processed at high temperatures [3]. Fructose is frequently present in fruits and honey. A food product with high fructose levels may be sweet and have a lot of natural sugars. Fructose content, which is frequently seen in processed foods, can also be an indication that the product has been sweetened with high fructose corn syrup [4]. Dietary carbohydrates contain xylose. Fruits, cereals, bread, and vegetables, including potatoes, peas, and carrots, all include it as part of their sugar composition. Detecting xylose can be used to identify the presence of specific types of fruits or vegetables in a food product. One way to recognize the presence of specific fruits and vegetables in a food product is to look for the sugar xylose, which is frequently present in certain foods. It is possible to identify and confirm the composition of a food product by analyzing the xylose content in a sample.

A large number of people worldwide suffer from diabetes [5,6], and it can lead to significant complications, such as heart attack, kidney failure, and blindness [7,8]. Moreover, glucose metabolism anomalies might lead to various diseases and problems [6,9]. Thus, it is highly crucial to monitor glucose levels and to supervise patients with diabetes. Researchers have developed various types of biosensors to help patients track their glucose concentration without the need to go to the hospital [10].

Blood sugar levels detection is a highly critical research area [11,12]. The two main electrochemical glucose identification methods are non-enzymatic or enzymatic [13]. Enzymatic methods utilize detection elements such as glucose oxidase enzyme (GOx). These methods oxidize glucose and generate compounds that can be detected, such as CO_2_, O_2_, or H_2_O_2_. When glucose oxidase interacts with enzymatic sensors, it releases oxygen, gluconolactone, and hydrogen. The glucose present on the sensor’s surface oxidizes and is expressed in typical current values. These sensors have high selectivity, however, they still face some challenges, such as: (i) degraded sensitivity with time due to enzymatic leaking; (ii) short life-time and low stability; and (iii) reduction in high overpotentials [14].

Until this time, most of the available blood glucose sensors depend on glucose oxidase (GOx) enzyme-based recognition unit [11,15]. Enzymatic glucose-based sensors have remarkable selectivity and sensitivity; however, they have low detection limits and are not stable with temperatures and humidity variations. Additionally, they require costly enzymes. Currently, research work is being focused on glucose biosensors that are cheaper, non-enzymatic, and sensitive to detect body glucose by using various body fluids, such as saliva, tear, or sweat for the purpose of self-monitoring of diabetes [12].

Various popular techniques can be used to detect glucose such as colorimetric and fluorometric methods [16,17,18,19,20,21]. The main idea behind these methods is using a chemical indicator or a fluorescent probe that changes colour or fluorescence intensity due to the addition of glucose target [21]. These methods are suitable for point-of-care applications as they are relatively simple, easy to use, and cheap. However, they lack the sensitivity of other techniques as the colour change or fluorescence signal can be influenced by other factors, such as temperature, pH, and the presence of other analytes.

One more popular method to detect glucose is paper-based sensors, which is also known as lateral flow assays [22]. Paper-based methods consist of a strip of paper coated with a reagent that is sensitive toward glucose. The targeted glucose sample is added to the strip, and the presence of glucose is identified by the change in colour and fluorescence [22,23]. These devices are popular as they are easy to use, portable, cheap, and simple. However, their signal can be changed due to the presence of other analytes and they are not as sensitive as other techniques [24].

Recently, there has been an increasing interest in electrochemical sensors [25] and biosensors to detect glucose since they provide high selectivity and sensitivity. Other various methods are being explored for the potential of glucose detection such as Raman spectroscopy [26] and mass spectrometry [27].

Electrochemical biosensing is applied extensively to detect biomolecules to diagnose and detect various diseases [28,29]. The biosensing research field has witnessed huge enhancement due to the development of field effect transistors biosensors. These biosensors have shown great performance due to their reliable detection, high sensitivity, and real time monitoring [30]. The sensing mechanism of transistor-based sensors depends on the change in the channel electrical resistance due to molecular addition and adsorption [31,32,33]. These devices have shown effective identification of molecules, ions, bacteria, and several biological entities [28,31,34,35,36,37].

As the biosensor performance relies on its surface to enhance the charge transfer, two-dimensional graphene including functionalized graphene nanomaterial is considered the best selection for glucose sensors. Platinum-functionalized graphene was utilized to detect glucose with a 0.6 M detection limit [38]. Moreover, gold nanoparticles were explored to detect 0.3 μM concentration of glucose [39]. Several nanomaterial biosensors, such as graphene and carbon nanotubes, were used for glucose detection. However, it poses the challenge of potential toxicity [40,41]. Various technologies were studied to design electrochemical reaction sensors based on non-enzymatic glucometers, including carbon-based materials, such as reduced graphene oxide (GO), graphene, metal nanoparticles [42,43], and carbon nanotube (CNT) [44,45].

Carbon nanomaterials doped with nitrogen have better performance in biosensors compared to pristine carbon. Carbon nanomaterials doped with nitrogen are used in biosensors because of their special characteristics that make them suitable for utilization in these kinds of applications [46]. Because the surface to volume ratios of carbon nanomaterials, such as carbon nanotubes and graphene are high, a lot of biomolecules can be adsorbed onto the surface [47]. Nitrogen atom doping of carbon nanomaterials improves their electrical conductivity, increasing their sensitivity for sensing applications [48]. It is possible to create nitrogen-doped carbon nanomaterials by adding nitrogen to carbon nanomaterials. These materials are more stable and have better electrical conductivity than pristine carbon. The electrical conductivity of carbon nanomaterials can be improved by nitrogen atoms acting as electron acceptors, increasing their sensitivity for biosensing applications [49]. Additionally, nitrogen doping can increase the carbon nanostructures’ chemical stability, strengthening their resistance to degradation. Carbon nanomaterials that have been doped with nitrogen are less toxic and more stable in biological settings, which can increase their biocompatibility. Additionally, compared to pristine carbon, nitrogen-doped carbon nanomaterials have demonstrated enhanced stability and biocompatibility, making them appropriate for application in biosensors [50]. Overall, nitrogen-doped carbon nanomaterials are a desirable option for use in biosensors due to their large surface area, electrical conductivity, and biocompatibility [46,48,50].

The novelty of this work is based on using C_2_N-FET for the first time as a sensor to recognize each of the sugar molecules. To the best of our knowledge, this is the first research that utilizes FET consisting of C_2_N channel and a pair of gold electrodes to identify glucose, fructose, and xylose molecules.

Within the many carbon nanostructures rich with nitrogen, C_2_N has been synthesized and computationally studied [51,52]. In this work, first principles modeling was used to study the sensing properties of C_2_N field effect transistor (FET) for the purpose of non-enzymatic glucose detection. This is the first report that uses C_2_N FET to detect glucose.

In this research, a field effect transistor based on two-dimensional nitrogenated holey graphene (C_2_N) was developed, designed, and tested to identify the sugar molecules including xylose, fructose, and glucose. Both density functional theory and non-equilibrium Green’s function (DFT + NEGF) were used to study the designed sensor. Various electronic characteristics were studied such as: work function, density of states, electrical current, and transmission spectrum. The proposed sensor is made of a pair of gold electrodes joint through a channel of C_2_N and a gate was placed underneath the channel. The C_2_N monolayer distinctive characteristics are promising for glucose sensors to detect blood sugar. Moreover, the detection of the three types of sugar molecules can be used to evaluate food quality.

## 2. Materials and Methods

The simulation work was produced using the graphical user interface of Virtual Nanolab and the Quantumwise Atomistix Toolkit (QuantumATK 2018.06 developed by Copenhagen, Denmark). United Arab Emirates University High Performance Computing (HPC) was utilized to run ATK-VNL simulations. Seven nodes with a total of 36 processors each have been used for HPC. As a result, 252 processors were used to complete the task.

### 2.1. Sensor Setup and Configuration

The setup and configuration of the C_2_N based sensor were conducted and investigated via Quantumwise (ATK-VNL). Figure 1 displays the nanoscale system setup. The left and right gold electrodes, the C_2_N central area which consists of one layer of C_2_N, and the gate terminal located beneath the central region make up the C_2_N metal-semiconductor-metal junction system. The gate is formed of two layers: a metallic layer and a 2.9 Å dielectric layer of SiO_2_ with a dielectric constant of 3.9. The C_2_N channel width is 13 Å and length is 28 Å, while the gold electrode length is 10 Å. The system consists of 209 atoms. First-principle electronic transport measurements were generated to detect each of the sugar molecules electronic signature. A, B, and C are indictors for A-, B-, and C-direction as displayed in Figure 1.

Figure 2 shows the atomic structure for each of the sugar molecules: glucose, fructose, and xylose. Due to their unique electronic and chemical structure, each molecule has a distinct electronic signature. Various electronic transport characteristics, including device density of states, transmission spectrum, work function, and electronic current, are generated for the bare C_2_N transistor and for the transistor with each of the sugar molecules. Figure 3 shows the C_2_N transistor structures with fructose. The big hollow site shown in Figure 3, which is the most stable site for xylose, fructose, and glucose for the adsorption of each of the sugar molecules [53]. The gate voltage was fixed at 1V, and finite bias voltage was fixed between right and left electrode and ranged from 0 to 1 V.

### 2.2. Computational Method

First-principles method is conducted within the generalized gradient approximation (GGA) exchange correlation function. For the plane-wave basis set, a cut-off energy of 80 Ha is utilized.

A 1 × 1 × 1 k-mesh is used to optimize the structure, while a denser mesh of 2 × 2 × 135 is used for the electronic transport calculations. The systems are optimized till the forces on each atom in the supercell are less than 0.05 eV/Å.

Each of the sugar molecules was optimized separately. Moreover, the gold atoms were optimized before forming the electrodes. Then, the C_2_N channel was optimized. At the end, the whole sensor with each of the sugar molecules was optimized. 1 × 1 × 1 k-mesh and Monkhorst-Pack grid, a type of uniform grid that is known to provide good convergence, were used for optimization as conducted by previous studies [54].

For the electronic transport characteristics such as IV a denser k-mesh grid was used. Quantumatk website [55] and other articles [56] recommend using 100 along the transport direction which is represented as the C direction in Figure 1. Thomas et al. used 1 × 1 × 100 k-point samplings along the device transport direction to generate the IV calculations [57]. In this work, a 2 × 2 × 135 k-point was utilized.

The electronic transport characteristics are generated by utilizing the density functional theory and non-equilibrium Green’s function (NEGF) approach. The sugar molecules are positioned on the C_2_N monolayer to investigate the transport characteristics of the C_2_N monolayer and the sugar molecules. Three areas are included: the left electrode, the right electrode, and the scattering region with each of the sugar molecules. The k-point grid for the electrodes and the scattering region calculation is 2 × 2 × 135.

The computed transmission probability of the electrons with energy (*E*) is generated, as shown in Equation (1):(1)$$\;\;T\left(E\right)=T_{r}\left\{\Gamma_{R}\left(E\right)\xi^{R}\Gamma_{L}\left(E\right)\xi^{A}\left(E\right)\right\}\;$$

Here, $\Gamma_{L}\left(E\right)$ and $\Gamma_{R}\left(E\right)$ are the broadening matrix for the left and right electrodes, respectively. $\xi^{A}$ and $\xi^{R}$ refer to the advanced and retarded Green’s function, respectively.

The zero bias conductance is generated with the relation $\xi =\xi_{0}T\left(E_{F}\right)$, where $\xi_{0}=2e^{2}/h$ is the quantum conductance. *E* and *h* refer to the electron charge and Planck’s constant, respectively.

The difference of the Fermi functions is used to calculate the integration of $T\left(E,V\right)$ over the energy window $f_{S,\;D}\left(E\right)={\left\{1+\exp\left[\left(E-E_{F}-eV_{S,D}\right)/k_{B}T\right]\right\}}^{-1}$, which gives the total current displayed in Equation (2):(2)$$I=\frac{2e}{h}\overset{\infty }{\underset{-\infty }{\int }}dE\;T\left(E,V\right)\left[f_{S}\left(E\right)-f_{D}\left(E\right)\right]$$

QuantumATK generates the density of state based on the following equations [58]:

The DeviceDensityOfStates (DDOS) $D\left(E\right)$ is computed via the spectral density matrix $\sigma \left(E\right)=\sigma^{L}\left(E\right)+\sigma^{R}\left(E\right)$, where *L*/*R* refers to the left and right electrodes.

The local density of states (LDOS) is computed as:(3)$$D\left(E,r\right)=\underset{ij}{\sum }\sigma_{ij}\left(E\right)\emptyset_{i}\left(r\right)\emptyset_{j}\left(r\right)$$

The basis set orbitals $\emptyset_{i}\left(r\right)$ are real functions in QuantumATK through the use of solid harmonics.

The device density of state is then obtained by integrating LDOS over all space:(4)$$D\left(E\right)=\int drD\left(E,r\right)=\underset{ij}{\sum }\sigma_{ij}\left(E\right)S_{ij}$$
where, $S_{ij}=\int \emptyset_{i}\left(r\right)\emptyset_{j}\left(r\right)dr$ is the overlap matrix. Introducing $M_{i}\left(E\right)=\sum_{j}\sigma_{ij}\left(E\right)S_{ij}$, the equation can be written as
(5)$$D\left(E\right)=\underset{i}{\sum }M_{i}\left(E\right)$$
where $M_{i}\left(E\right)$ is considered as the contribution of DDOS from orbital *i*. $M_{i}\left(E\right)$ is a spectral Mulliken Population with:(6)$$M_{i}=\int M_{i}\left(E\right)f(\frac{E-\mu }{k_{B}T})\;dE$$

## 3. Results and Discussion

The electrical transport properties were generated for the C_2_N FET to achieve the practical investigation of the designed C_2_N FET sensor to specifically detect each of the sugar molecules. Density of states, work function, transmission spectrum, current variation, and current-voltage characteristics were generated for the C_2_N FET, the C_2_N FET with the presence of glucose molecule, the C_2_N FET with the presence of fructose molecule, and for the C_2_N FET with the presence of xylose molecule.

### 3.1. Device Density of States (DDOS)

A distinct and significant change in the FET Device DOS have been noticed in the presence of the different sugar molecules. Figure 4 displays a comparison of the DDOS for the bare C_2_N FET (without any target molecule) and for the C_2_N FET in the presence of each of the sugar molecules. Figure 4a shows that the bare C_2_N FET have more energy states than the C_2_N FET in the presence of glucose molecule, which can be observed at the energy levels of −3.8, −3.6, −3.2, and −2.9 eV. Furthermore, the presence of fructose molecule affected the C_2_N FET DOS differently, as displayed in Figure 4b, where a new energy spike can be observed at energy level 3.85 eV. Similarly, a significant change in DOS can be noticed in the C_2_N FET when it is exposed to xylose molecule, as displayed in Figure 4c. Two new energy spikes were noticed at energy levels of 3.7 and 3.9 eV, as shown in Figure 4c.

Figure 5 displays the partial DOS, which reflects a closer look and more detailed information about the effect of each of the sugar molecules on the DDOS. It was noticed that, when a target molecule is added to the device, one unique peak is increased in the DDOS due to glucose (Figure 5a) or fructose (Figure 5b) or xylose (Figure 5c). This indicates that adding each of the sugar molecules results in new electronic states within the energy range of that peak. This may indicate that the sugar molecule is interacting with the C_2_N channel and modifying its electronic structure. The change in the DDOS is caused by the sugar molecule accepting or donating electrons from the channel material or by forming chemical bonds between the target molecule and the C_2_N channel.

The DOS of a material is defined as the measure of the number of available electronic states within a certain energy range. The DOS changes due to the presence of various types of molecules since they can introduce defects of impurities into the material, which leads to a change in the electronic structure of the material. As an example, when the material is exposed to a target molecule, the impurities can result in additional energy levels, which can modify the density of states. Moreover, impurities affect electronic states symmetry, which modifies the DOS. Additionally, the mechanical and chemical properties can be changed leading to a change in the DOS. The variation in the DOS depends on the type and concentration of the defects.

### 3.2. Work Function

The C_2_N FET response to each of the sugar molecules is investigated by calculating the work function displayed in Figure 6. The calculated work function value for the C_2_N FET is 5.92 eV; for the C_2_N FET with glucose, it is 6.08 eV; for the C_2_N FET with fructose, it is 6.05 eV; and for the C_2_N FET with xylose, it is 6.106.

Figure 6 shows an increment in the work function for the C_2_N FET with each of the sugar molecules in comparison to the bare C_2_N FET. This increment indicates that the adsorption of each of the sugar molecules leads to a decrement in the electron mobility. The work function increment is caused by the cloud charge transfers from the C_2_N channel toward the sugar molecules. The study’s findings are in line with previous research work [53].

The increment in the work function of C_2_N due to presence of each of the sugar molecules is believed to be associated with changes in the electronic characteristics of the C_2_N material due to the interaction between each of the target molecules and the C_2_N surface [53]. The energy needed to remove an electron from the surface can increase when the target molecule accepts electrons from the C_2_N material, increasing the work function.

Moreover, it is expected that the movement of charge carriers from the C_2_N material to the sugar molecules leads to a decrement in the density near Fermi level. Thus, the Fermi level shifts to higher energies leading to an increment in the work function.

### 3.3. Transmission Spectrum

Figure 7 shows the transmission spectra T(E) for the C_2_N FET with and without each of the sugar molecules (glucose, fructose, and xylose) at different biases: (a) V = 0 V, (b) V = 0.2 V, and V= 0.4 V. The figure shows the changes in transmission spectrum when different sugar molecules are added at a varying applied voltage. The transmission spectrum has a low value in the energy range [0.4, 0.9] eV because of the energy window within the band gap of the semiconducting C_2_N channel.

### 3.4. Current-Voltage

The current vs. voltage characteristics for the C_2_N FET sensor and for each of the sugar molecules adsorbed via the C_2_N FET sensor are shown in Figure 8. A fixed 1 V gate potential was used while the V_ds_ was set to 0.2, 0.4, 0.6, 0.8, and 1 V. Figure 8 shows the current voltage curves for C_2_N FET at 0.2, 0.4, 0.6, 0.8, 1 V before and after the addition of each of the sugar molecules. The variation in current reading with the addition of sugar molecules indicates successful detection. The adsorbed target molecule interacts with the C_2_N-FET and changes its conductivity by changing the carriers’ concentration. C_2_N is a semiconducting nanomaterial, which has a nonlinear resistance, resulting in a nonlinear IV curve, as shown in Figure 8.

The current of the C_2_N-FET differs noticeably for each sugar molecule. The size, electrical state, and way that each sugar molecule interacts with the C_2_N-FET channel are all unique. When the gate potential was fixed at 1 V and the bias voltage among the left and right electrodes was fixed at 0.4 V, the created sensor produced the best results. The best sensitivity was achieved by setting the bias voltage at 0.4 V, as shown in Figure 8. This work is a proof of concept that the developed C_2_N-FET can be utilized to detect the different types of sugar molecules.

The sensor showed the best sensitivity at 0.4 V bias voltage. Figure 9 shows the sensor’s response (change in current), where the highest variation in the electrical signal was due to glucose molecule adsorption. These results show that the device has high selectivity for glucose and results in a distinct electrical current for each of the sugar molecules. The current variation is due to the change in the charge and the electrical potential after introducing the target molecules which alters the charge carriers’ density. Thus, the sensor conductivity and current change.

This work is a proof of concept that the modeled and studied C_2_N FET can be utilized as a sensor for sugar molecules detection, such as glucose, fructose, and xylose. This research indicates that each of the sugar molecules have a unique electronic signature that can be identified via the designed C_2_N FET.

After employing the computational methods to detect each of the sugar molecules, the results of the sensor can be utilized to identify the performance in real-time applications. Such computational methods provide valuable results, such as how each of the sugar molecules will interact with the sensor. These results can be utilized to optimize the sensor design and performance in terms of stability, sensitivity, and selectivity. Moreover, the used computational method provides insights into the electronic transport characteristics of the system due to each of the sugar molecules. These electronic properties include electron density, work function, transmission spectrum, and current–voltage measurements. These results can be utilized to understand how the target molecule interacts with the sensor and affects the sensor’s performance.

After the validation of the sensor via computational methods, the sensor can be designed, fabricated, and tested in real-time applications. Then, the sensor’s performance can be evaluated by comparing the computational expectations with the experimental findings.

In this work, C_2_N FET was utilized to detect each one of the sugar molecules separately, where each one of them resulted in a unique electronic signature and unique variation in current indicating the possibility of detecting each of them in real-time applications. The highest sensitivity was toward glucose molecule, which can be used to monitor and control diabetes.

The addition of a mixture of two or three sugar molecules is also expected to result in a specific variation in current and a unique electronic signature, since each sugar molecule interacts with the C_2_N channel and modifies its electronic properties in a unique way.

In general, the employed computational method results in valuable information about the performance of the designed sensor. However, computational methods do not show the limit of detection of the sensor in real-time applications. The limit of detection can only be identified by experiment by measuring the response of the fabricated sensor toward various concentration of the target analyte.

Combining both computational method with experimental data can be used to overcome the limitations of such technology. This comparison leads to identifying the potential sources of error and uncertainty. It is worth mentioning that the employment of computational methods enables researchers to suggest future directions to study to enhance the sensor’s performance and then test it experimentally.

Introducing structure variables, such as surface roughness, pores, and alien molecules to a sensor, will result in a significant effect on its electronic properties and performance.

In terms of work function, the existence of surface roughness or defects lead to a shift in the work function. Moreover, surface roughness can also affect the amount of charge that can be stored on the device, which affects its sensitivity.

In terms of density of states, impurities and defects can generate localized states within the bandgap of the sensor, which can modify electrical current and the conductivity of the device. Moreover, surface roughness and pores can affect the DOS by creating additional pathways for charge carriers to pass through.

In terms of current, impurities and defects work as scattering centers for the charge carriers, which might lead to a reduction in the device mobility and current. Introducing structure variables impact the electronic properties and performance of a sensor, affecting its work function, density of states, and current.

## 4. Conclusions

Real-time identification of the different sugar molecules is essential for monitoring and preventing diabetes and to evaluate food quality. In this research, a field effect transistor based on two-dimensional nitrogenated holey graphene (C_2_N) was designed, developed, and tested to identify the sugar molecules, including xylose, fructose, and glucose. To investigate the characteristics of this device, non-equilibrium Green’s function and density functional theory (NEGF + DFT) were utilized. Various electronic properties were studied, including density of states, work function, transmission spectrum, and electrical current. The proposed sensor consists of a pair of gold electrodes connected via a channel of C_2_N and a gate. The electronic characteristics of the C_2_N FET changed because of the adsorption of the target molecules. The measurable variations in the electronic characteristics with each sugar molecule validate the potential of the C_2_N FET sensor in detecting sugar molecules. The C_2_N monolayer distinctive characteristics are promising for glucose sensors to detect blood sugar.

## Figures and Tables

**Figure 1 nanomaterials-13-00700-f001:**
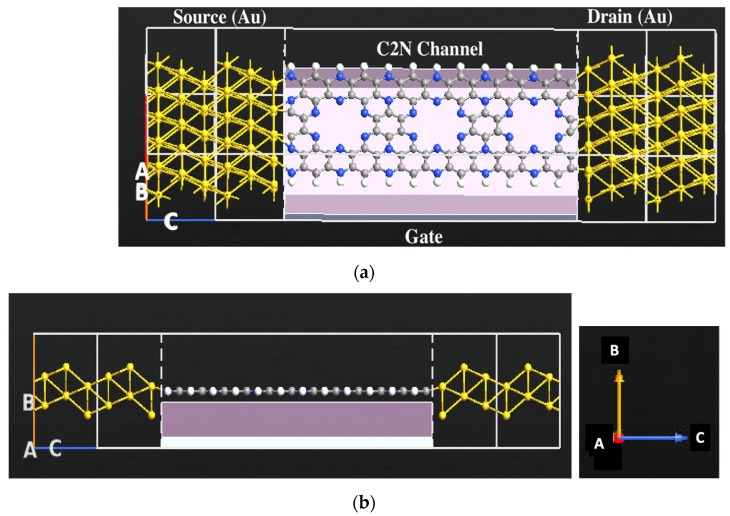
C_2_N-FET sensor designed by ATK-VNL. (**a**) Schematic representation of the C_2_N-FET device. (**b**) Cross-sectional view of the C_2_N-FET device. The built sensor consists of two gold electrodes, a monolayer C_2_N channel and a gate underneath the channel. Color code: hydrogen—white, nitrogen—blue, gold—yellow, and carbon—gray.

**Figure 2 nanomaterials-13-00700-f002:**
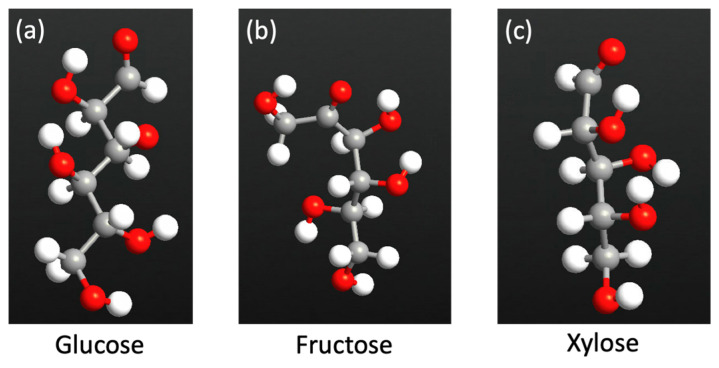
Atomic structure of each of the sugar molecules: glucose (**a**), fructose (**b**), and xylose (**c**). Color code: oxygen—red, carbon—gray, and hydrogen—white.

**Figure 3 nanomaterials-13-00700-f003:**
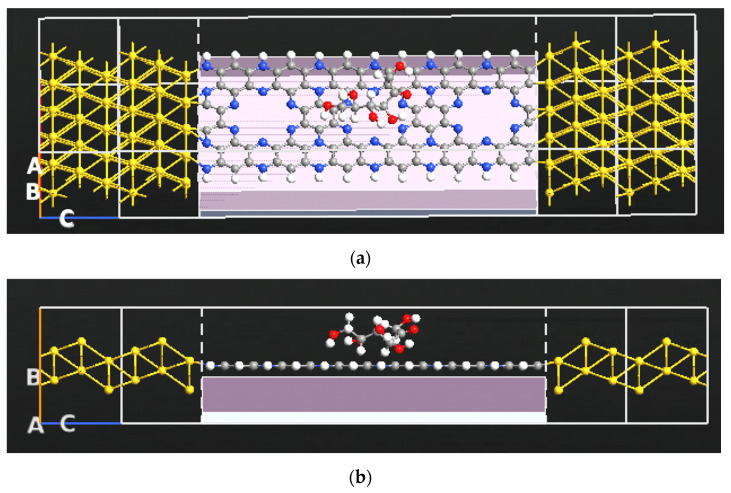
(**a**) Schematic diagram of the C_2_N-FET sensor with fructose. (**b**) Cross-sectional view of the C_2_N-FET sensor with fructose.

**Figure 4 nanomaterials-13-00700-f004:**
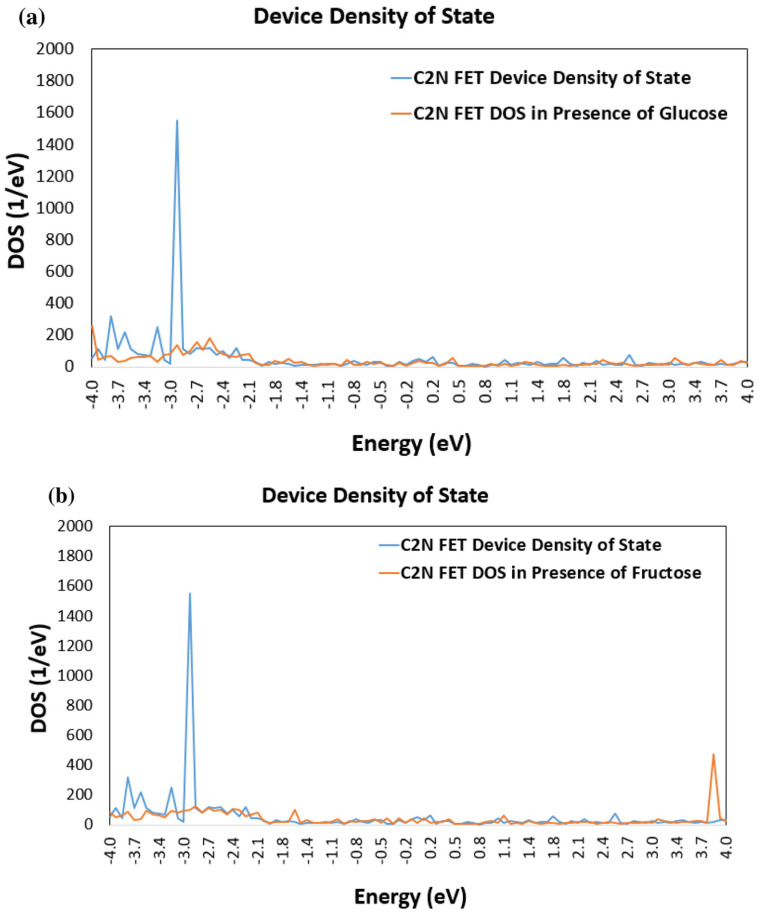
Change in device density of states (DOS) of simulated C_2_N FET in presence of (**a**) glucose molecule; (**b**) fructose molecule; and (**c**) xylose molecule.

**Figure 5 nanomaterials-13-00700-f005:**
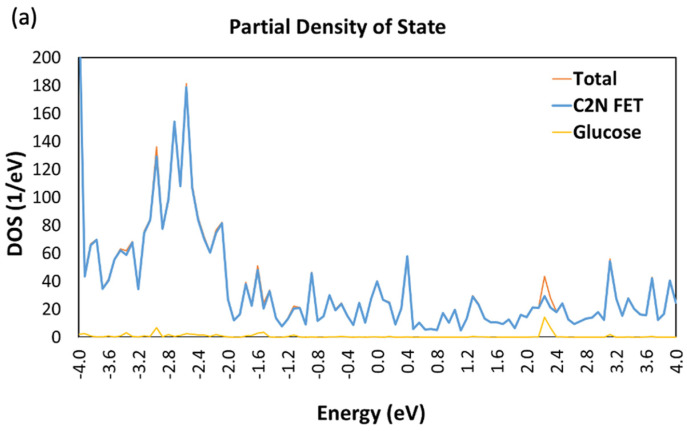
Total and partial density of states (DOS) of C_2_N FET in presence of (**a**) glucose molecule, (**b**) fructose molecule, and (**c**) xylose molecule.

**Figure 6 nanomaterials-13-00700-f006:**
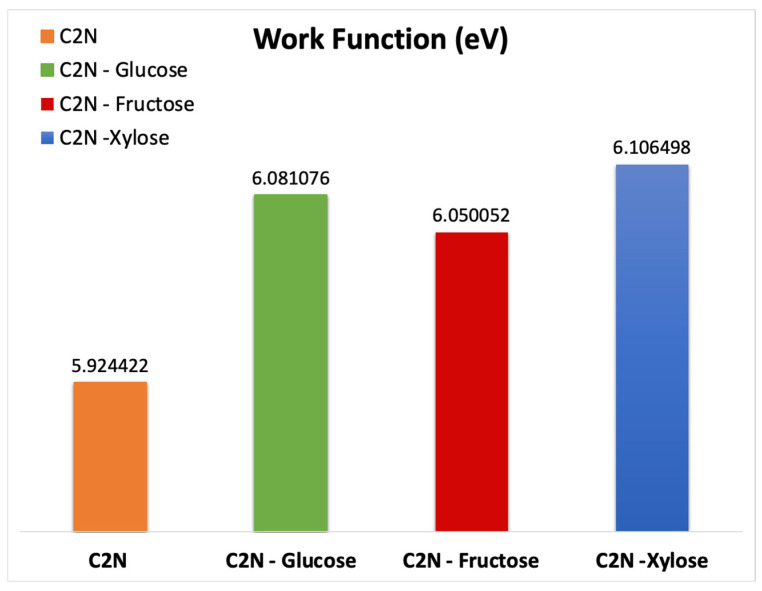
Work function of: C_2_N FET; C_2_N FET with the presence of glucose molecule; C_2_N FET with the presence of fructose molecule; and C_2_N FET with the presence of xylose molecule.

**Figure 7 nanomaterials-13-00700-f007:**
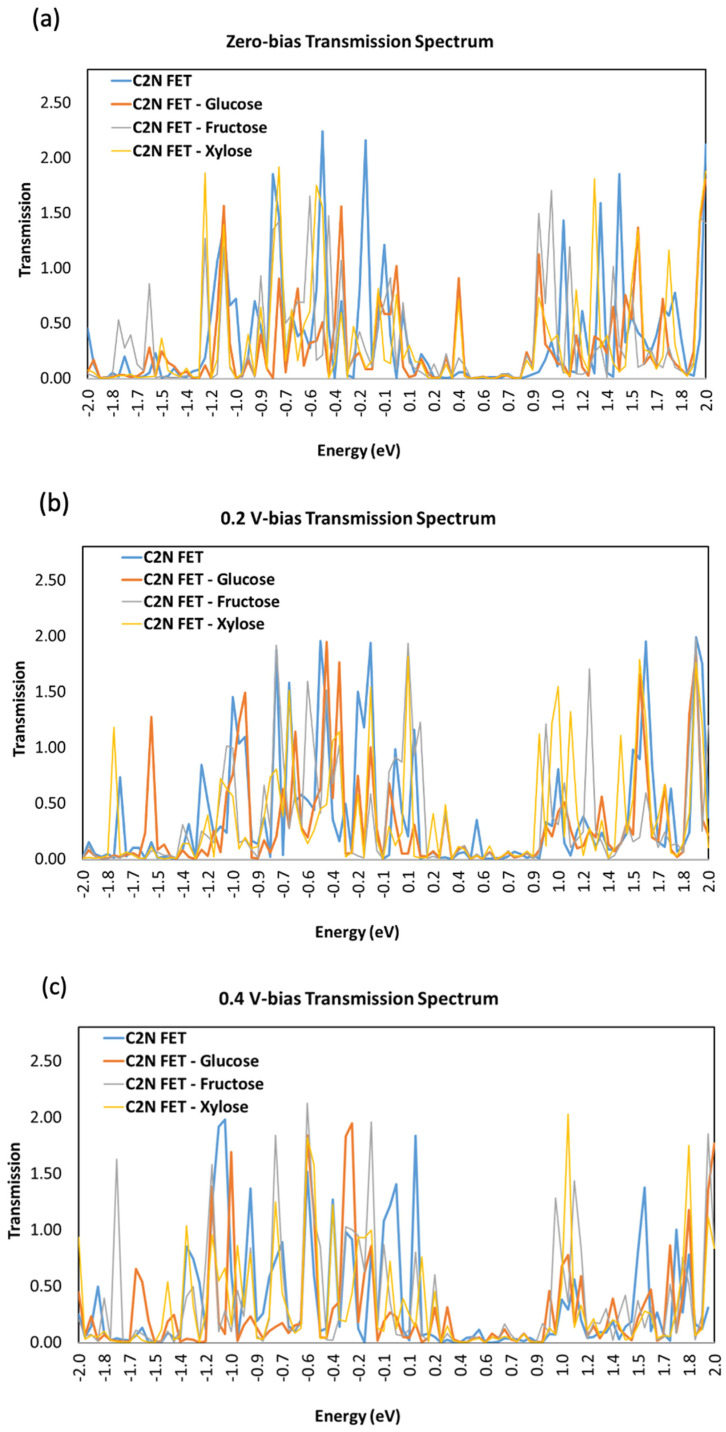
Transmission spectra T(E) for C_2_N-FET sensor with three different sugar molecules (**a**) bias voltage = 0 V, (**b**) bias voltage = 0.2 V, and (**c**) bias voltage = 0.4 V.

**Figure 8 nanomaterials-13-00700-f008:**
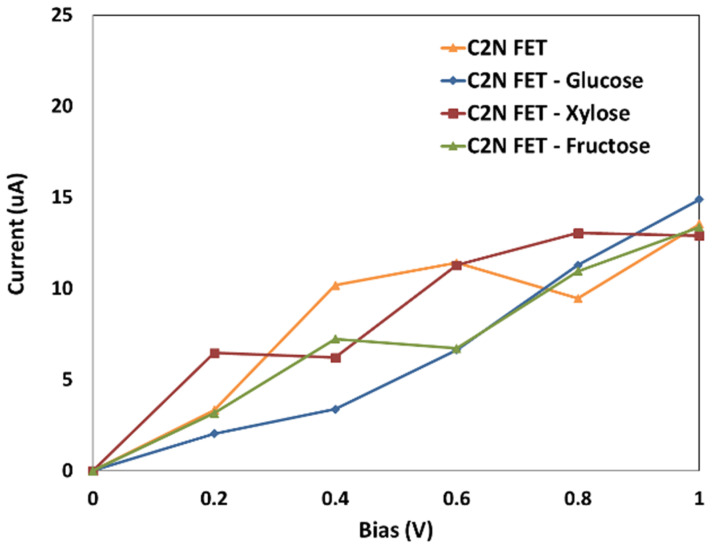
Current–voltage characteristics vs bias for the C_2_N FET (orange), for the C_2_N FET with glucose (blue), for the C_2_N FET with xylose (red), and for the C_2_N FET with fructose (green).

**Figure 9 nanomaterials-13-00700-f009:**
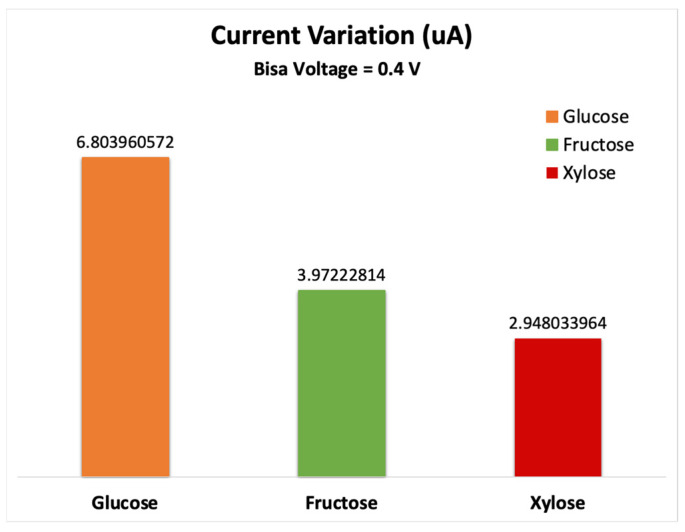
Variation in electrical drain current for the various types of sugar molecules.

## Data Availability

All data generated or analyzed during this study are included in this published article.

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
