# Peer review of "Sugar Molecules Detection via C2N Transistor-Based Sensor: First Principles Modeling"

_nanomaterials, 2023, doi:10.3390/nano13040700_

Round 1

Reviewer 1 Report

In this work, by employing first-principles modelling techniques, a field effect transistor (FET) based on two-dimensional Nitrogenated Holey Graphene (C2N) was designed, developed, and tested by Wasfi et al. to identify the sensing capabilities of sugar molecules including xylose, fructose, and glucose. This work is well presented and could generate some interest in many working in the field of 2D materials. So, I recommend this paper for publication after addressing some minor comments.

1)  In the introduction, the authors mentioned that “Carbon nanomaterials doped with nitrogen have better performance in biosensors compared to pristine carbon”. This is a general statement. The authors need to explain the motivation for selecting this material with the help of proper literature. The actual research gap is missing in this manuscript.

2)  Again, the authors mentioned that “the detection of the three types of sugar molecules can be used to 95 evaluate food quality”. How? Need explanation.

3)  The 1 × 1 × 1 k-mesh for optimization does not seem to be a good computational strategy. Did the author check the K-grid convergence before finalizing the simulation parameters? Similarly, the k-grid selection for electronic properties should be higher (something like 13x13cx1) for a 2D material. Please have a look at the paper https://doi.org/10.1021/acsanm.0c02072.

4) While explaining the DOS, the authors also need to include the contribution of each element or atom in the whole system as well as need to include the PDOS of the system. This will help to better understand the electronic properties of C2N in the bare form as well as with the presence of sugar molecules. 

5)    The increase of C2N work function with the presence of sugar molecules needs more explanation. Is it only associated with electron mobility?

6)    Some typos:

(a) Page 2: line 91 should be the density of states. This typo is repeated in many places and that needs to be corrected.

(b) Some of the Figure qualities are bad and the authors need to increase the legend size in most figures. 

Author Response

Manuscript title: Sugar Molecules Detection via C2N transistor-based Sensor: First-Principles Modeling.

Manuscript number: Nanomaterials-2174953

Dear Editor,

We are grateful to the reviewers for their valuable comments, which helped enhance the quality of our manuscript. Our responses to the reviewer's comments are shown in blue text in the attached document and the changes are highlighted in the manuscript.

Thank you & best regards,

Falah Awwad

Reviewer 2 Report

Asma et al. developed a simulated strategy to detect sugar molecules such as glucose, xylose, and fructose. C2N-based FET was designed to test the efficiency. However, the manuscript lacks to address the real-time applicability of the setup. There is no significant explanation of the possibility of real-time device development and it's not clear to the reviewer. There are a few concerns regarding the manuscripts listed below for authors. The manuscript needs to be improved further to qualify the expectations of the Journal nanomaterials.   

1.       Line no. 38, please rewrite the sentence “……….to evaluate fruit quality such as fruit, fruit juice, and honey.’ This is somehow not clear in delivering the actual meaning.

2.       It is also advised to include brief input on trends in other strategies such as colorimetric/fluorometric, and paper-based devices for glucose detection in the introduction section.

3.       The manuscript explains computational methods for sugar molecule detection. However, it is not clear what will the outcome of the device for real time applications? Please justify.

4.       It is not clear what is the change in DOS, work function, current-voltage characteristics, and current variations of simulated C2N FET in presence of a mixture of three sugar molecules and a mixture of two sugar molecules. Justify.

5.       What is the limit of detection of the developed technology for each sugar molecule? How do authors address the limitations of the technology? How to overcome the limitations?

Author Response

(The authors gave the same response as above.)

Reviewer 3 Report

Overall, the authors conducted a modeling work based on a field effect transistor on two-dimensional Nitrogenated Holey Graphene (C2N). The model sensor was developed, designed, and tested to identify the sugar molecules including xylose, fructose, and glucose. Density functional theory and Nonequilibrium Green's function (DFT+ NEGF) were used to study the designed sensor’s properties. Various electronic characteristics were studied such as: work function, density of state, electrical current, and transmission spectrum. The proposed sensor is made of a pair of gold electrodes joint through a channel of C2N and a gate was placed underneath the channel. The C2N monolayer distinctive characteristics are promising for glucose sensors to detect blood sugar. The flow of the paper is smooth. However, many details need to be added for this modeling work. The comments are shown as following:

Comment 1: In line 84, FET should be written with its full name.

Comment 2. What do the A, B, and C represent in Figure 1? Please illustrate. Also, in the model, only a layer of C2N is built or several layers of the material are stacked together?

Comment 3. How is the density of state (DOS) calculated based on the proposed equation 1 and 2?

Comment 4: Can you explain why the DOS shows such a big difference when exposed to different molecules? As shown in Figure 4?

Comment 5: Overall, the modeling is reasonable. The whole system is based on an easy and ideal system without any structure defects or pores. Can the author discuss what the results will be if defects and structure variables (surface roughness, pores, and alien molecules) are introduced?

Author Response

(The authors gave the same response as above.)

Round 2

Reviewer 2 Report

The authors have addressed all the comments. However, there are few minor confusions in the manuscript.

1.       Line no. 50, the sentence ‘Xylose can be used to identify the presence of specific types of fruits or vegetables in a food product’. Its not clear. Please check.

2.       The work is at the stage of proof of concept, reviewer believes the scope of journal does not match well with the manuscript. There is a need to do real-time experimentations to fully realize the potential of the sensor.

3.       The manuscript needs to be further improved to match the standard of the journal. 

Author Response

Response to the comments:

  • Referee 2:

The authors have addressed all the comments. However, there are few minor confusions in the manuscript.

We thank the reviewer for the good impression of our manuscript. The updated version of the manuscript has been enhanced with the reviewer's recommendations.

  1. Line no. 50, the sentence ‘Xylose can be used to identify the presence of specific types of fruits or vegetables in a food product’. It’s not clear. Please check.

Thanks for the comment.

Addressed, and the text below was amended on page 2.

Detecting xylose can be used to identify the presence of specific types of fruits or vegetables in a food product. One way to recognize the presence of specific fruits and vegetables in a food product is to look for the sugar xylose, which is frequently present in certain foods. It is possible to identify and confirm the composition of a food product by analyzing the xylose content in a sample.

  1. The work is at the stage of proof of concept, reviewer believes the scope of journal does not match well with the manuscript. There is a need to do real-time experimentations to fully realize the potential of the sensor.

Thank you for your valuable feedback and comments on our manuscript. We understand your concerns regarding the proof-of-concept stage and the need for real-time experiments to fully realize the potential of the sensor.

We would like to emphasize the significance of our work in developing the field and offering new perspectives on the design and application of sensor technology. The novelty of this work is based on the use of C2N-FET for the first time as a sensor to recognize each of the sugar molecules. To the best of our knowledge, this is the first research that utilizes an FET consisting of a C2N channel and a pair of gold electrodes to identify glucose, fructose, and xylose molecules.

Our proof-of-concept stage demonstrates the viability and potential of our proposed strategy and provides a strong platform for further development and testing. We believe that the findings and insights presented in our manuscript are valuable contributions to the field. Our proof-of-concept study provides a thorough analysis of the C2N FET and offers significant insights into its potential for real-world applications. The results of our research show that our proposed strategy is viable and pave the way for future research and development.

  1. The manuscript needs to be further improved to match the standard of the journal. 

Thank you for your feedback. We acknowledge that the manuscript needed improvement since its first submission to this journal to meet its high standards. We have worked diligently to make the necessary revisions and enhance the quality of the content to meet the requirements. Thank you again for your input.
